# Melatonin Ameliorates Neuropsychiatric Behaviors, Gut Microbiome, and Microbiota-Derived Metabolites in Rats with Chronic Sleep Deprivation

**DOI:** 10.3390/ijms242316820

**Published:** 2023-11-27

**Authors:** Bingcong Li, Yin-Ru Hsieh, Wen-De Lai, Te-Hsuan Tung, Yu-Xuan Chen, Chia-Hui Yang, Yu-Chiao Fang, Shih-Yi Huang

**Affiliations:** 1School of Nutrition and Health Sciences, Taipei Medical University, Taipei 110301, Taiwan; bingcong@umich.edu (B.L.); ma07111011@tmu.edu.tw (Y.-R.H.);; 2Graduate Institute of Metabolism and Obesity Sciences, Taipei Medical University, Taipei 110301, Taiwan; 3Nutrition Research Center, Taipei Medical University Hospital, Taipei 110301, Taiwan; 4TMU Research Center for Digestive Medicine, Taipei Medical University, Taipei 110301, Taiwan

**Keywords:** melatonin, sleep deprivation, depression-like behavior, cognitive, intestinal metabolites

## Abstract

With the increasing prevalence of sleep deprivation (SD)-related disorders, the effective treatment of sleep disorders has become a critical health research topic. Thus, we hypothesized and investigated the effectiveness of a 3-week melatonin intervention on neuropsychiatric behavioral responses mediated throughout melatonin receptors, gut microbiota, and lipid metabolites in rats with chronic SD. Eighteen 6-week-old Wistar rats were used and divided into the control grup (C, n = 6), SD group (n = 6), and melatonin-supplemented group (SDM, n = 6). During weeks 0 to 6, animals were provided with the AIN-93M diet and free access to water. Four-week chronic SD was conducted from weeks 7 to 10. Exogenous melatonin administration (10 mg/kg BW) was injected intraperitoneally 1 h before the daily administration of SD for 3 weeks in the SDM group. SD rats exhibited anxiety-like behavior, depression-like behavior, and cognitive impairment. Exogenous melatonin administration ameliorated neuropsychiatric behaviors induced by chronic SD. Analysis of fecal metabolites indicated that melatonin may influence brain messaging through the microbiota–gut–brain axis by increasing the production of short-chain fatty acids (SCFA) and decreasing the production of secondary bile acids (SBA). Four-week SD reduced the cerebral cortex expression of MT1, but not in the colon. Chronic SD led to anxiety and depression-like behaviors and cognitive decline, as well as the reduced intestinal level of SCFAs and the enhanced intestinal level of SBAs in rats. In this work, we confirmed our hypothesis that a 3-week melatonin intervention on neuropsychiatric behavioral response mediated throughout melatonin receptors, gut microbiota, and lipid metabolites in rats with chronic SD.

## 1. Introduction

The National Sleep Foundation reported that 7 to 8 h of sleep daily is essential for good health, but most adults in the United States have sleep deprivation (SD) [1]. The Centers for Disease Control and Prevention also reported that health problems caused by SD incur hundreds of billions of dollars in healthcare costs and over $400 billion in lost Gross Domestic Product (GDP) yearly [1,2]. Many causes, such as poor sleep hygiene, lifestyle choices, work patterns, sleep disorders, or other medical conditions, have been attributed to decreased sleep time and abnormal physical and mental status [3]. Chronic SD, cardiovascular disease, poor immune function, high risk of metabolic disease, and abnormal body hormone status exhibit strong associations [4,5,6,7]. Sleep deprivation also affects the mental state and mental health, with poor sleep having a high positive correlation with mental disorders such as depression, anxiety, and even bipolar disorder [8].

Furthermore, individuals with mental health problems are more likely to have insomnia or other sleep disorders [9]. According to neuroimaging studies, chronic SD may be a leading cause of negative thinking and emotional vulnerability [10]. A meta-analysis review also demonstrated that higher sleep quality enhances cognitive function, including memory, problem-solving, creativity, emotional processing, and judgment [11]. Sleep restriction overworks brain neurons and is attributed to poor individual performance in thinking, decision-making, memory, and cognition [12]. Because treating or alleviating SD can enhance mental and emotional resilience and improve cognitive and behavioral impairments caused by SD, identifying the active substances to improve the mental health problems caused by SD is a crucial topic.

Melatonin (N-acetyl-5-methoxytryptamine) is a hormone synthesized and secreted by the pineal gland at night [13]. The supraoptic nucleus regulates the secretion rhythm of melatonin, and melatonin is secreted in response to the light–dark cycle of light. In the darkness, the pineal gland secretes melatonin; it inhibits melatonin production in the daytime [14,15]. Melatonin binds to neuron melatonin receptors 1 and 2 (MT_1_ & MT_2_) to trigger these biological actions. MT_1_ is involved in activating the PI3K/Akt signaling, contributing to the growth, proliferation, and survival of neurons, and MT_2_ contributes to the cAMP/PKA/CREB signaling pathway to regulate the expression of the circadian rhythm genes [16]. A meta-analysis demonstrated that melatonin significantly reduces sleep onset latency and increases the total sleep duration [17]. Exogenous melatonin has also been shown to cause significant improvements in older patients with insomnia and patients with Alzheimer’s disease and sleep disorders [18]. Animal studies have reported a decline in the expression of MT_1_ and MT_2_ in the hippocampus under conditions of acute SD, which may be attributable to the overactivation of autophagy and apoptosis in neural cells and may result in cognitively impaired behaviors [19] and the dysregulation of melatonin receptor expression in patients with depression and insomnia [20]. Thus, recovering the expression of melatonin receptors might be a strategy for alleviating depressive symptoms in patients with insomnia.

Research has revealed causal relationships between the gut microbiome and mental behaviors and has elucidated the underlying molecular mechanisms [21,22,23]. A potential pathway through which the gut microbiota interacts with the host is through the metabolites of the gut microbiota, such as short-chain fatty acids (SCFAs), secondary bile acids (SBAs), and indoles [24]. These metabolites may act as signals that affect host immune constancy, energy homeostasis, and maintenance of the architectural integrity of the intestinal epithelium [25,26]. SCFAs are considered neural messaging molecules and microbiota-gut-brain axis messengers [25]. SBAs, especially taurine-conjugated bile acids, significantly correlate with behavioral outcomes in rats with chronic unpredictable mild stress [27]. However, the relationship between the bile acid–gut microbe–disease and its mechanisms remains unclear. Thus, we hypothesized and investigated the effectiveness of a 3-week intervention of melatonin on neuropsychiatric behavioral responses mediated throughout melatonin receptors, gut microbiota, and lipid metabolites in rats with chronic SD.

## 2. Results

### 2.1. Growth Status and Organ Weights of Animals

Before the SD period (weeks 0–6), the rats exhibited a steady increase in body weight, with no difference in weight gain between the groups. After the start of SD at week 7, the SD and SDM groups began to exhibit lower body weights than the C group. During the SD treatment period (weeks 8–10), the body weight of the SD and SDM groups steadily increased; however, their final body weight remained lower than that of the C group (Figure 1).

As shown in Table 1, the relative weights of perirenal and epididymal fat were significantly lower in the SD and SDM groups than in the C group (*p* < 0.05). As shown in Table 2, the relative weight of the cerebral cortex was significantly higher in the SD and SDM groups than in the C group (*p* < 0.05).

### 2.2. Biochemical, Inflammatory, and Neurochemical Status

We observed significant decreases in serum TG, free fatty acid (FFA), and BUN levels in the SD and SDM groups compared with the C group (all *p* < 0.05; Table 2). Serum levels of TNF-α and IL-1β were significantly higher in the SD group than in the C group (*p* < 0.05). The pro-inflammatory effect of SD on serum lipopolysaccharide levels was significantly reversed after melatonin supplementation (*p* < 0.05; Table 2). Our results demonstrated that the rats with SD secreted more corticosteroids than those in the C group (*p* < 0.05). By contrast, serum corticosteroid levels were significantly lower in the SDM group than in the SD group. No differences in serum β-NGF or dopamine levels were observed among the groups (Table 2).

### 2.3. Behavioral Responses

In this study, chronic SD increased the anxiety-like behavioral responses of the rats in the OFT and EPM. In the OFT, the SD and SDM groups had significantly shorter central zone visit times and travel distances and significantly fewer central zone visits than the C group (*p* < 0.05). The central zone visit times, central zone travel distances, and central zone visits of the SDM and SD groups were similar. In the EPM, the SD group spent significantly less time in and had fewer visits to the open arm in the EPM than the C group (*p* < 0.05). By contrast, the SDM group spent significantly more time in and had significantly more visits to the open arm in the EPM than the SD group (*p* < 0.05; Table 3). To examine the depression-like behavior of the rats, we performed the FST during the 4 weeks of chronic SD induction. The SD group exhibited significantly longer immobility times than the C group (*p* < 0.05). Immobility times were significantly shorter in the SDM group than in the SD group (*p* < 0.05; Table 3). Similarly, the SD rats exhibited a lower preference for sucrose water at week 10 in the SPT than the other rats.

In the MWM, significant differences were observed in the effects of melatonin and SD treatments. The interaction of group × learning day on escape latencies among the groups in MWM acquisition training are shown in Figure 2a (*p* < 0.05). The SDM group exhibited a significantly lower escape latency than the control group during the 1st training day (*p* < 0.05; one-way ANOVA, Figure 2b). During the 3rd training day, the SD group required more time to reach the platform than the control group; however, no differences were observed in the escape latency between the groups (*p* > 0.05; Figure 2a).

### 2.4. Status of Gut Microbiota and Its Metabolites

We examined the status of gut microbiota under SD and melatonin supplementation. The results of the β-diversity analysis presented distinct clustering of the colonic microbiota composition in the three groups (Figure 3a,b). To identify specific bacterial taxa associated with SD and melatonin supplementation, we employed the LEfSe method. Melatonin supplementation significantly suppressed some pathogenic bacteria as well as decreased the ratio of Firmicutes and Bacteroidetes (F/B ratio). A cladogram representative of the colonic microbiota structure indicated the predominant bacteria and differences in taxa among the three communities (Figure 4a,b). The results showed that the predominant bacteria at the genus level in the SD rats were Negativibacillus, Ruminiclostridium, and Raoultella. The predominant bacteria in the SDM rats were Blautia, Christensenellaceae, Lachnospriaceae, Lactococcus, Candidatus Stoquefichus, and Fecalitalea; however, the predominant species were not similar to those observed in the control group. In brief, these results indicated that melatonin supplementation reversed SD-induced colonic microbiota dysbiosis.

The levels of SCFAs in fecal samples are presented in Figure 5. We observed and labeled SCFAs containing straight or branched-chain fatty acids with two to six carbon atoms. Fecal butyric acid concentrations did not differ between the SD and C groups. However, the SD group exhibited significantly lower fecal concentrations of acetic acid, propionic acid, isobutyric acid, and total SCFAs than the C group (*p* < 0.05 for all). The fecal concentrations of acetic, propionic, isobutyric acid, and butyric acid did not differ between the SD and SDM groups. However, the fecal concentrations of total SCFAs were significantly higher in the SDM group than in the SD group.

We also investigated the effect of melatonin supplementation on the SBAs metabolized by the gut microbiota. We measured the fecal concentrations of SBAs. The fecal concentrations of total SBAs were higher in the SD group than in the C group. After 4 weeks of SD induction, the levels of selected SBAs increased, especially those of iso-lithocholic acid (LCA) and iso-deoxycholic acid (DCA) (both *p* < 0.05; Figure 6b,d). The concentrations of DCA significantly decreased under melatonin supplementation. However, the SDM group did not show any significant change in total SBAs, namely 12-keto-LCA, LCA, iso-LCA, and hyodeoxycholic acid (HDCA) compared to the SD group (Figure 6).

### 2.5. Cerebral Cortex and Colon Melatonin Receptor Protein Expression Levels

According to Figure 7, compared with the control group, the expression of MT_1_ proteins in the cerebral cortex and colon was significantly elevated in the rat groups with CSD. Melatonin administration did not alter the expression levels of MT_1_ and MT_2_ in both cerebral cortex and colon tissues. Correlation analysis was conducted to elucidate the interconnection between alterations in the protein expression levels and neuropsychiatric behaviors of rats. The results revealed that the protein expression level of MT_1_, but not that of MT_2_, in the cerebral cortex was significantly positively associated with total distance traveled and central visit time in the OFT and negatively with immobile time in the FST (Table 4).

## 3. Discussion

This work aimed to investigate the effectiveness of melatonin on the alleviation of chronic SD-induced anxiety and depression-like behaviors and cognitive decline in rats. Such neuropsychiatric behavioral responses might be attributed to the alternations of gut microbiota, lipid metabolites, and expression of melatonin receptors. In this work, we confirmed our hypothesis that a 3-week melatonin intervention on neuropsychiatric behavioral responses mediated throughout melatonin receptors, gut microbiota, and lipid metabolites in rats with chronic SD. Exogenous melatonin supplementation significantly improved anxiety behaviors and cognitive decline in rats with chronic SD and corrected abnormalities in intestinal metabolism.

The present study assessed the effect of exogenous melatonin supplementation on SD in rats. The dose and timing of the melatonin supplementation intervention were determined based on preliminary screening of the literature [28,29]. Traditionally, interventional melatonin is given orally, but the bioavailability is low in this mode; review studies have collated the pharmacokinetics of various melatonin administration routes. Intraperitoneal administration is associated with rapid absorption rates and high bioavailability. Moreover, by avoiding first-pass metabolism, intraperitoneal administration is more clinically relevant. Combining the absorption rate, metabolic effect, and practical factors led to the use of intraperitoneal injection as the optimum administration route for melatonin intervention [30]. Due to the quick metabolism of melatonin, we chose intraperitoneal administration to reveal the optimal function of melatonin in the chronic sleep deprivation rat model. More dosage of oral administration of melatonin might be expected in patients with sleep disorders.

In this study, the specific gravity of the perirenal and epididymal tissues was lower in the SD and SDM groups than in the C group. Studies have suggested that SD causes the inflammation of the white adipose tissue, in which the increased secretion of the cytokine IL-6 induces lipolysis, resulting in the decreased weight of the white adipose tissue [31]. The results of this study revealed that the serum concentrations of TG and FFA were significantly lower in the SD group than in the C group. Studies have suggested that the environmental stress induced by SD may accelerate the anabolic metabolism of fat, resulting in decreased serum concentrations of circulating fatty acids such as TG and FFA [32]. Higher levels of corticosterone secreted under chronic stress may be critical to the catabolic status of subjects and may result in lower blood lipid levels, more significant adipose tissue loss, and weight loss under SD.

Regarding the relationship between SD and liver and kidney function, this study identified no difference in the ratios of liver function indicators (alanine transaminase, aspartate transferase, and total bilirubin) to nutritional indicators (albumin) and in the related indicators in the serum between the C and SD groups. Based on biochemical data, no safety concern was identified during the 3-week intraperitoneal melatonin administration. The chronic SD pattern used in this study did not cause liver damage in the rats, probably because the deprivation was not intense and was administered over a long period, providing the rats with enough time to adapt to the new rest and sleep pattern. Regarding the serum renal function index, the BUN levels in the SD and SDM groups were significantly lower than those in the C group; however, the values were in the normal range. SD has been demonstrated to increase renal oxidative stress, causing renal function impairment and edema [33]. The biochemical values observed in this study indicated an increase in the renal body weight ratio, and we presume that chronic SD may cause kidney damage.

Melatonin improved SD-induced anxiety behaviors in animals in an elevated cross-maze experiment. By contrast, melatonin altered SD-induced anxiety behavior in an open field test. Clinical studies have demonstrated that melatonin supplementation can be used to treat sleep problems effectively. Many studies have investigated the other indications of melatonin, including improving sleeping disorders in anxiety animal models. Melatonin may improve anxiety by improving sleep. Melatonin can increase γ-aminobutyric acid (GABA) concentrations in specific brain areas, producing a sedative effect and reducing anxiety symptoms in a rat model [34]. The present study also revealed that melatonin improved anxiety-like symptoms in an animal model of chronic SD.

In this study, the EPM and FST results suggest that melatonin supplementation did improve SD-induced depressive behavior in the rats. The pharmacological capability of melatonin to regulate the sleep cycle makes it a suitable daily supplement for improving and treating various sleep problems. However, whether melatonin can improve SD-induced depressive-like behavior is unclear. Reynolds et al. reported that individuals with depression produce more melatonin in the brain at night [35]. A clinical review also reported that melatonin might slightly improve depressive symptoms but without a significant effect [36]. Patients with anxiety disorders often have difficulty sleeping in clinical practice and attribute this to insomnia [37]. In this study, we observed that chronic SD caused anxiety-like behaviors in the OFT and EPM rats. Melatonin supplementation ameliorated anxiety-like behavior in the rats in the EPM but not in OFT. Melatonin supplementation can improve anxiety symptoms in patients before surgery [38]. Melatonin supplementation may enhance GABA levels in selected brain regions, and these higher GABA concentrations can produce sedation and reduce anxiety [34,39].

Endogenous melatonin secretion is regulated by the physiological clock and light, which inhibit melatonin synthesis and directly promote long-term memory formation. By contrast, the peak melatonin concentrations at night inhibit memory consolidation [40]. Melatonin administration improves spatial learning and memory in SD animals in the MWM [41]. Exogenous melatonin supplementation can improve cognitive function in older adults [42] and individuals with chronic SD and various chemically induced cognitive impairments [43]. Higher physiological melatonin concentrations are significantly correlated with a lower prevalence of cognitive impairment, depressed mood, and Alzheimer’s disease [44,45].

Primary intestinal lipid metabolites include SCFAs and SBAs. Propionic and butyric acid are the most physiologically active SCFAs. In this study, we detected lower SCFA production under chronic SD; however, the melatonin intervention nonsignificantly enhanced SCFA levels. Physical, psychological, and dietary status may alter the gut microbiota profile and metabolite production [46]. A previous study reported that the abundance of SCFA production–related microbiota, *Blautia*, *Candidatus*, *Fecalitalea*, and *Lactococcus*, was elevated through a dietary intervention under chronic SD [47]. SCFAs might be vital to maintaining the integrity of the gut barrier and the transduction of gut–brain axis communication [48]. In the present experiment, unlike the intestinal metabolites of the SD rats, the intestinal bacteria of the SDM rats tended to metabolize more SCFAs and fewer SBAs. We found significantly higher fecal concentrations of total SBAs in the SD group than in the SDM group, and they were negatively correlated to depression-like behaviors, poor cognitive responses, lower SCFA levels, and higher inflammatory cytokine levels. High concentrations of SBAs cause the increased permeability of epithelial cells, making it easier for harmful substances to enter the circulation and cause damage to peripheral tissues and even the central nervous system. Mahmoudian Dehkordi et al. reported that microbial dysbiosis, which produces excessive SBAs, may be a significant cause of cognitive changes in patients with Alzheimer’s disease [49]. In summary, 3-week melatonin administration elevated fecal SCFA concentrations and reduced fecal SBA concentrations. This finding indicates that the melatonin intervention might be involved in the alteration of the status of the gut microbiota and their metabolites, which is attributed to the improvement of anxiety and depressive-like symptoms and cognitive capability under chronic SD.

Alteration of gut microbiota has been reported to be modified by diets, medicine, and abnormal metabolic statuses [50,51,52]. The microbiota compositions of the mice, especially in terms of the relative abundances of *Bacteroidetes*, Lachnospiraceae and Bifidobacterium, responded to the rapid dietary changes between fish oil and high-fat diet [53]. Researchers have proposed that n-3 PUFAs directly affect gut microbial diversity and alter the metabolic function of the host [27,54]. Our previous study [47] demonstrated that chronic SD altered the richness and diversity of gut microbiota. In this study, we observed a decrease in the diversity and richness of the gut microbiota after 4 weeks of chronic SD, especially a significantly increased abundance of harmful bacteria at the genus level, such as *Ruminiclostridium*, *Candidatus*, and *Negativibacillus* in the SD rats. Both *Negativibacillus* and *Candidatus* are Gram-negative species and may produce endotoxins. *Ruminiclostridium* was significantly increased in the intestinal flora of animals with acute necrotizing pancreatitis and diarrhea [55]. Both the SCFA and SBA of the gut microbiota and the gut microbiota have been suggested to play significant roles in mental health through the gut–microbiota–brain axis [56]. Gut bacteria-produced SCFAs and SBAs have been reported to trigger the signal-provoking potential actions from the gut to the brain [57]. An increased circulating LPS concentration was noted in this depressive-like-induced SD study. This phenomenon might be attributable to the increased levels of pathogenic bacteria, alterations in intestinal flora, and elevation in bacterial metabolites. The Firmicutes/Bacteroidetes ratio tended to be higher in the SD group than in other groups. Another study also reported that a high-fat diet affects the host neuroendocrine system and gut microbiota ecology and is a critical factor in depressive-like behavior development [58].

The expression levels of MT_1_ and MT_2_ in the brain vary under chronic SD. Melatonin receptors are involved in regulating the circadian rhythm and are a therapeutic target in depressive disorder [59]. In a melatonin receptor-knockout model, MT_2_ regulated the nonrapid eye movement sleep time, and MT_1_ was involved in determining the rapid eye movement sleep time [60]. In this study, 4-week SD increased the expression of MT_1_, but not that of MT_2,_ in the cerebral cortex; however, this did not differ under melatonin administration. Cao et al. reported that the MT_1_ protein level in the hippocampus decreased in a 72-h acute SD rat model [61]. Regardless of the SD type, the enhanced expression of melatonin receptors and the increased levels of GABA in selected brain regions may contribute to sedation and reduced anxiety [34].

In this study, we also evaluated the melatonin receptor levels in the colon. Colonic MT_1_ and MT_2_ expression increased significantly under chronic SD. Pandi-Perumal et al. reported that pineal gland and intestinal enterochromaffin-like cells could secrete melatonin; however, the melatonin levels in the colon and feces, but not in the serum, decrease significantly under SD [62]. In a pinealectomy rat model, MT_1_ maintained its regulation of the circadian rhythm. Colonic melatonin secretion and receptors may have other physical functions besides those in circadian rhythm regulation [63]. In a study, decreased the activation of nuclear factor kappa B expression regulated intestinal melatonin levels. This was attributed to the enhanced oxidative status and inflammation that resulted in lower intestinal melatonin secretion and increased melatonin receptor expression under SD [64].

Some limitations of this study should be noted. First, the chronic SD-induced depression-like rat model has been used to mimic the depression status in some murine species; however, in this study, the physical status of the rats cannot be controlled for this behavioral design. Other models, such as the unpredictable chronic mild stress model or the modified multiple platform water bath method, should be considered to eliminate the possibility that the rats were recognized, exhausted, and over-alert after chronic SD. Second, the number of samples used for gut microbiota and its metabolite analyses was limited and insufficient to elucidate possible mechanisms underlying microbiota involvement in neurodegeneration development. Although the SCFA and SBA were analyzed in this study, the levels of SCFA and other possible lipid metabolites in the blood and brain should be considered. Few studies have investigated the effect of melatonin on lipid metabolism, especially in the brain. Although further studies are required to investigate the relationship between CSD and differentially expressed lipid species because most of the restored lipid species in the SR rats were significantly correlated with anxiety- and depressive-like behavior test indices, we speculate that an interaction occurred between the membrane lipids and melatonin receptors. More studies should be conducted to elucidate the effects of dietary lipids on the composition of microbiota, especially in clinical trials.

In conclusion, chronic SD led to anxiety and depression-like behaviors and cognitive decline, as well as the reduced intestinal level of SCFAs and the enhanced intestinal level of SBAs in rats. In this work, we confirmed our hypothesis that a 3-week melatonin intervention on neuropsychiatric behavioral response mediated throughout melatonin receptors, gut microbiota, and lipid metabolites in rats with chronic SD.

## 4. Materials and Methods

### 4.1. Animals and Diets

Wistar male rats (BioLASCO Taiwan, Taipei, Taiwan) aged six weeks and weighing 300–350 g were used in this study. They were housed in an environment with constant and controlled humidity and temperature under a 12-h light-dark cycle (light on 7 a.m.~7 p.m.). After 2-week acclimatization (fed with Laboratory Rodent Diet 5001), the eighteen rats were divided into three experimental groups: a control group (C, n = 6), an SD group (n = 6), and a melatonin-supplemented group (SDM, n = 6) (Figure 1). During weeks 0 to 6, all animals were provided with the American Institute of Nutrition-93M diet (AIN-93M, MP Biomedicals, Santa Ana, CA, USA) and free access to water. The dietary composition of the experiment diet is presented in Table 1. Four-week chronic SD was conducted from weeks 7 to 10. The rats’ food intake and body weight were recorded during the experiment. This animal study was approved by the Animal Care and Use Committee of Taipei Medical University (LAC-2018-0499).

### 4.2. Melatonin Intervention

In this experiment, for the SDM group, melatonin at a dose of 10 mg (#M5250; Sigma-Aldrich, USA) per kg of body weight was injected intraperitoneally 1 h before the daily administration of SD for three weeks (from weeks 8 to 10). Before injection, it was dissolved in 0.5 mL of anhydrous ethanol and 15 mL of saline. The C and SD groups were injected with 15 mL of saline containing 0.5 mL of anhydrous ethanol.

### 4.3. Chronic Sleep Deprivation

The rats were subjected to chronic SD using a modified multiplatform water bath for 28 consecutive days from 4 p.m. to 10 a.m. the following day. Eight transparent columnar platforms were placed in a water bath. Rats from SD and SDM groups were placed in the water bath to induce chronic SD. The water surface was 1 cm below the platform’s top, allowing the rat to return to the platform to avoid drowning quickly. When the rats reached the rapid eye movement stage of sleep, the muscles lost tension, causing the rats to fall into the water. The rats then awoke and attempted to climb onto the platform to avoid drowning. Throughout the experiment, the water in the tank was replaced with clean water daily. In addition, the electric heaters were set up around the tank to prevent the rats from dying of hypothermia after falling into the water.

The C and SD groups were fed a regular diet. In the SDM group, melatonin at a dose of 10 mg (#M5250; Sigma-Aldrich, USA) per kg of body weight was injected intraperitoneally for 3 weeks. The SD and SDM groups were subjected to sleep deprivation from 4 p.m. until 10 a.m. for 28 days using a modified multiplatform water bath.

### 4.4. Behavioral Tests

Anxiety-like behaviors in the rats were measured using the open field test (OFT) and the elevated plus maze (EPM). The modified OFT was conducted to measure the anxiety-like behaviors in rats [65]. The apparatus consisted of a 50 × 50 × 40 cm^3^ quadrilateral black box. The open field was divided into a central 20 × 20 cm^2^ main area rounding periphery. Recorded videos were analyzed with ActualTrack (Actual Analytics, Edinburgh, Scotland) to measure the total distance traveled by the rats in 5 min and the number of entries into and time spent in the central area. A study validated the use of the EPM to examine the anxiolytic effects of pharmacological compounds and elucidate the mechanisms of anxiety-related behavior [66]. The apparatus in this study consisted of four arms that were 550 cm long, 110 cm wide, and 50 cm high. A black wall 20 cm in height enclosed the closed 20 cm; the rats were placed in the central area of the four arms in the maze, facing an open arm. Entries and the duration spent in each arm were recorded for 5 min using ActualTrack (Actual Analytics, Edinburgh, Scotland).

Depressive-like behaviors were measured using the sucrose preference test (SPT) and forced swim test (FST) [67]. After 24-h water deprivation, each animal was exposed to two bottles for 1 h, one containing deionized water and another containing 1% sucrose solution (*w*/*w*). The positions of the two bottles were varied randomly. The sucrose preference percentage was calculated using the following formula: sucrose solution intake/total intake, where total intake = sucrose solution intake + water intake, as previously described [47]. The consumption of sucrose solution was an indicator of depression-like behavior in each group.

In the present experiment, we used a modified FST procedure [68]. A rat was placed in a vertical acrylic cylinder (70 cm in height and 35 cm in diameter) containing 20 °C ± 2 °C water at a depth of 35 cm. The day before the formal experiment, the rat was placed in water for a 15-min pretest; 24 h later, the rat was tested for 5 min under the same experimental conditions, and videos were recorded (Sony, Japan). The videos were analyzed using Forced Swim Scan 2.0 (CleverSys, Reston, VA, USA) to measure the immobility and struggle (swimming, climbing, escaping, and diving) times for each rat. The rats with longer immobility times and shorter struggle times were considered to exhibit depression-like behavior.

In the present experiment, the rats’ cognitive–spatial learning and memory were using the Morris water maze (MWM) [69]. The apparatus consisted of an open circular black pool (100 cm in height and 150 cm in diameter) approximately half filled with water (20 °C ± 2 °C), a hidden platform in a fixed position under the water surface, and extra maze cues on the walls of the room. Before the test, the rats were trained four times daily for 3 days. During the 3-day learning period, the time to reach the platform was measured as an indicator of learning. On day 4 of the probe test, each rat was allowed to swim for 40 s. The time spent in the target quadrant was recorded and used to indicate the memory level in the probe trials. The behaviors of the rats were tracked and recorded using CineLyzer (Plexon, Dallas, TX, USA).

### 4.5. Examination of Biochemical, Inflammatory, and Neurochemical Parameters

After the behavioral tests, the rats were anesthetized, and blood was collected from the abdominal artery, immediately placed into tubes, and stored on ice for 30 min. After centrifugation at 3000 rpm for 10 min at 4 °C, the top layer of plasma and serum was removed and stored at −80 °C.

Kidney function index (blood urea nitrogen [BUN] and creatinine), liver function index (glutamic oxaloacetic transaminase and glutamic pyruvic transaminase), triglyceride (TG), total cholesterol, low-density lipoprotein cholesterol, and high-density lipoprotein cholesterol were measured in the serum samples. The biochemical parameters were measured using a clinical chemistry analyzer (UniCel DxC 800, Beckman Coulter, Brea, CA, USA). Pro-inflammatory parameters (tumor necrosis factor [TNF]-α and interleukin [IL]-1β) were measured in partial plasma samples. Plasma TNF-α levels were examined using an enzyme-linked immunosorbent assay (ELISA) kit specific for TNF-α in rats (Cat. No. 438207, BioLegend, San Diego, CA, USA) as per the manufacturer’s instructions. Serum IL-1β levels were determined using a quantized ELISA kit specific for IL-1β in rats (Cat. No. RLB00, R&D Systems, Minneapolis, MN, USA). Levels of corticosteroid, nerve growth factor (NGF), and dopamine in partial plasma samples were examined using ELISA (Rat NGF/NGF-β ELISA kit, Abcam, Cambridge, UK; Dopamine kit, Labor Diagnostika Nord, respectively) as per the manufacturer’s instructions. Six plasma samples were measured for each group.

### 4.6. Collection of Colon Content and Fecal Microbiota Analysis

Fecal samples from the rats were collected at 10 a.m. every week and before the rats were sacrificed. The collected samples were immediately frozen in liquid nitrogen. Subsequently, the fragments were stored in a −80 °C refrigerator until further analysis.

The DNA was extracted from 200-mg fecal samples using the PowerSoil DNA Isolation Kit (Qiagen, Hilden, Germany) under the manufacturer’s instructions. The 16S amplicon sequencing method was used to amplify the 16s ribosomal RNA (rRNA), and an Illumina MiSeq system (San Diego, CA, USA) was used to sequence the 16s rRNA genes extracted from the fecal samples. Five fecal samples were analyzed in each group. We demultiplexed the raw data (Illumina CASAVA v1.8), merged the paired-end reads, and removed any chimeras (UCHIME). AAUPARSE was used to cluster operational taxonomic units (OTU), with a similarity cut-off of 97%. The inferred amplicon sequence variants were subjected to taxonomy assignment using the SILVA database (v132) with minimum bootstrap confidence of 80%. The community diversity was estimated based on the normalized reads using the Shannon-Wiener and Simpson’s diversity indexes. Partial least squares discriminant analysis (PLS-DA) and permutational multiple analysis of variance (MANOVA) were conducted to identify significant differences between samples. Linear discriminant analysis effect size (LEfSe) was used to identify features that are statistically different among bacterial taxonomy with the Kruskal–Wallis test to detect features with significant differential abundance with respect to the class. Detailed 16s ribosomal RNA sequencing raw data of gut microbiota are given in Appendix A.

### 4.7. Measurement of Microbiota-Derived Metabolites

The fecal samples obtained from the rats subjected to 4-week SD were used to analyze acetic, propionic, and isobutyric acid concentrations. Butyric acid was detected using gas chromatography, a method modified from that of Zhao et al. [70]. In brief, the feces (100 mg) were mixed with 0.5% phosphate solution (1 mL) and homogenized for 20 s. The homogenates were centrifuged for 10 min at 14,800 rpm. Then, 200 μL of the supernatant was mixed with the same volume of ethyl acetate and centrifuged for 10 min at 14,800 rpm. The supernatant was stored at −80 °C until further analysis. The analysis was conducted using a gas chromatograph (Agilent 7820/5977B GC-EI MSD) that was equipped with a flame ionization detector and fitted with a capillary column (internal diameter: 30 m × 0.25 mm, film thickness: 0.25 μm; Nukol Capillary GC column, Merck KGaA, Darmstadt, Germany). The helium pressure was 30 kPa, the injector temperature was 250 °C, the detector temperature was 230 °C, and the injection volume was 1 μL. The oven conditions were as follows: 1 min at 90 °C, increased to 150 °C at 15 °C/min, increased to 170 °C at 5 °C/min, increased to 250 °C at 20 °C/min, and held at 250 °C for 2 min. Peaks were identified using the retention time. The internal standard peak area ratio was determined for each SCFA peak, and the corresponding SCFA concentration was determined from calibration curves. Agilent MassHunter Workstation Software was used for analysis. The prevalence and concentration of each SCFA were included as variables in the analysis. The feces (10 mg) were mixed with an internal standard solution (500 μL; DCA-d6, GCA-d4, and TCDCA-d4) and extracted using acetonitrile and formic acid mixture. The supernatant was used for bile acid analysis, which was performed using an ultrahigh-performance liquid chromatography system with Xevo TQS MS (Waters). Chromatographic separation was performed using an ACQUITY BEH C8 column (2.1 mm × 100 mm × 1.7 μm, Waters). The column temperature was maintained at 60 °C. Regarding optimized parameters, mobile phase A was 10% acetonitrile with 0.01% formic acid, and mobile phase B was isopropanol/acetonitrile (50:50, *v*/*v*) with 0.01% formic acid. Mass analysis was performed using the Xevo TQ-S system (Waters) in the positive ion electrospray ionization mode. The capillary voltage was set at 1.5 kV. The desolvation gas flow rate was set at 1000 L h^−1^, and the cone gas flow rate was maintained at 150 L h^−1^. The desolvation and source temperatures were set at 600 °C and 150 °C, respectively. The quality control sample (laboratory quality control) and mixed quality control sample (a mixture of all models) were prepared for analysis during analytical runs after every tenth sample. TargetLynx was used to determine signal intensity integration and concentration conversion.

### 4.8. Western Blotting

The rat cerebral cortex and colon tissue were prepared in RIPA buffer (RB4475, Bio Basic, Markham, ON, Canada) supplemented with a protease inhibitor cocktail (P8340, Sigma-Aldrich, CA, USA) and were centrifuged for 30 min at 14,000 rpm. Transfer the supernatant to a new tube and perform Bradford protein assay. 40 μg protein was electrophoresed on 10% SDS-PAGE and then transferred onto polyvinylidene difluoride membranes. All membranes were blocked using 5% bovine serum albumin solution, and primary antibody incubation was performed overnight at 2 °C to 8 °C. The membranes were incubated with secondary antibodies for 1oneh and visualized using the UVP BioSpectrumAC Imaging System (Analytik Jena, Jena, Germany) with enhanced chemiluminescence reagent (#WBBLUF0500, Merck Millipore, Burlington, MA, USA). The primary antibodies used were Melatonin Receptor 1A antibody (1:1000, #bs-0027R, Bioss, Woburn, MA, USA), Melatonin Receptor 1B antibody (1:1000, #ab203346, Abcam, Waltham, MA, USA), and anti-β-actin monoclonal antibody (1:5000, #T0022, Affinity Biosciences, Cincinnati, OH, USA). The secondary antibodies used were Anti-mouse IgG, HRP-linked Antibody (1:5000, #7076S, Cell Signaling, Danvers, MA, USA) and Anti-rabbit IgG, HRP-linked Antibody (1:5000, #7074S, Cell Signaling, Danvers, MA, USA).

### 4.9. Statistical Analysis

The data were analyzed using Prism 8 statistical software (GraphPad). Differences between the groups were examined using analysis of variance with Tukey’s multiple comparison tests. The data were presented as mean ± standard error, and *p* < 0.05 was considered a statistically significant difference among the groups. Pearson’s correlation coefficient was used to analyze the correlation between melatonin receptor expression level and neuropsychiatric behaviors. Specific statistical details pertinent to fecal microbiota analysis are presented within the entire methodological parts.

## Figures and Tables

**Figure 1 ijms-24-16820-f001:**
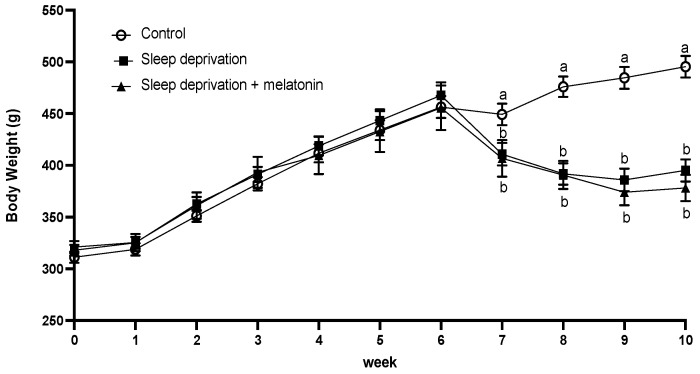
Body weight changes in chronically sleep-deprived rats. Values are presented as mean ± standard error (n = 6). Different letters indicate significant differences among groups at *p* < 0.05 as per one-way analysis of variance.

**Figure 2 ijms-24-16820-f002:**
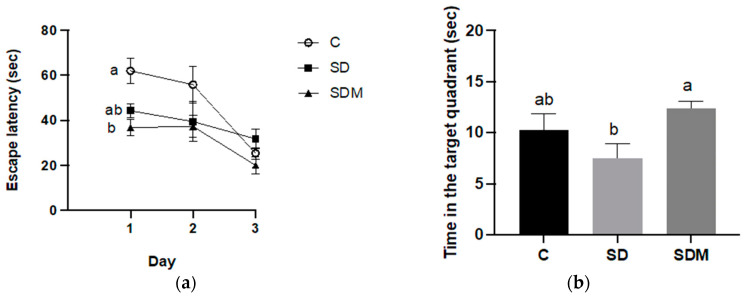
Melatonin administration alters the behaviors of Morris water maze performance in sleep-deprivation rats. (**a**) Escape latency of 3-day acquisition trial for each trial in a session of four training trials; (**b**) Time spent in the target quadrant during probe trial on day 4. Values are presented as mean ± standard error (n = 6). Different letters indicate significant differences among groups at *p* < 0.05 as per one-way analysis of variance. C—control; SD—sleep deprivation; SDM—sleep deprivation + melatonin.

**Figure 3 ijms-24-16820-f003:**
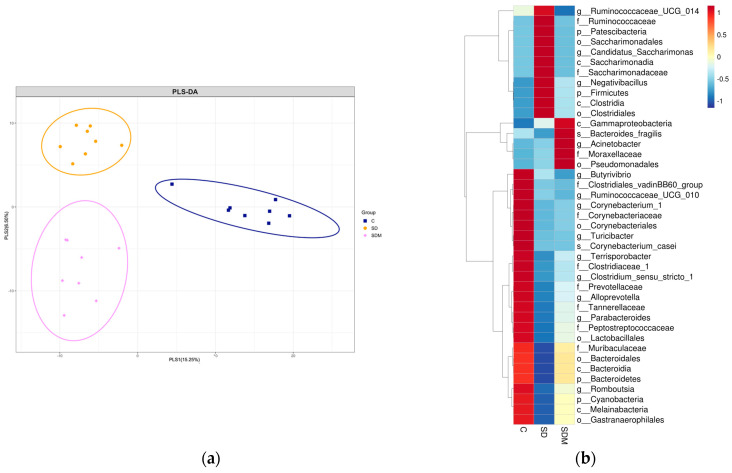
Melatonin administration on (**a**) β-diversity of gut microbiota community (n = 6) and (**b**) heatmap of microbiota relative abundance (n = 6) in sleep-deprived rats. (**a**) Partial least square discriminant analysis (PLS-DA) score plot based on the relative abundances of OTU in the gut microbiota and their association with interventions. The axes indicate the percentages of variation in the data for the bacterial communities. Dot symbols in different representing colors that cluster together are circled in blue, orange, and pink, respectively. (**b**) Heatmap of microbiota relative abundance. The redder the color, the higher the relative abundance; the bluer the color, the lower the relative abundance. C—control; SD—sleep deprivation; SDM—sleep deprivation + melatonin.

**Figure 4 ijms-24-16820-f004:**
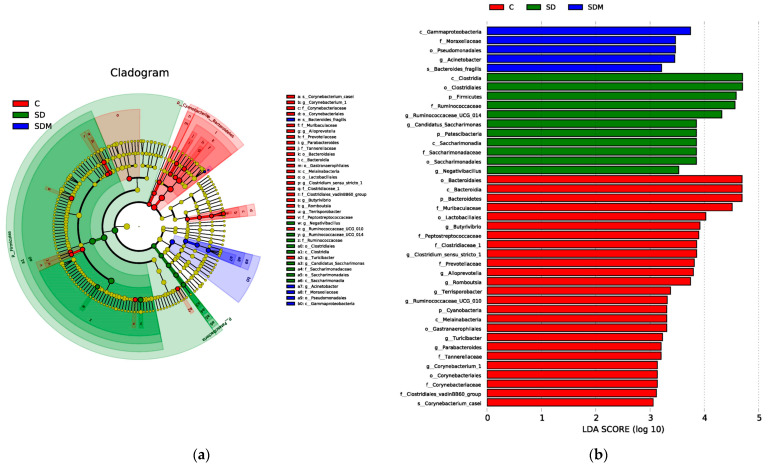
Melatonin administration on abundant bacterial taxa (n = 6) in sleep-deprived rats. Cladogram representation (**a**) of differentially abundant bacterial families detected using linear discriminant analysis effect size (LEfSe). Different colors indicated the group in which clade was most abundant. Significant bacterial genera (**b**) were determined by the Kruskal–Wallis test (*p* < 0.05) with an LDA score greater than 3. C—control; SD—sleep deprivation; SDM—sleep deprivation + melatonin.

**Figure 5 ijms-24-16820-f005:**
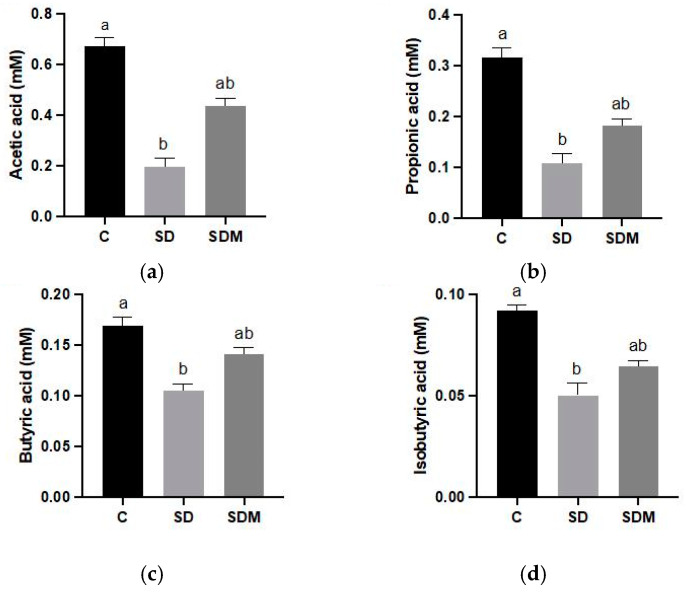
Melatonin administration on fecal short-chain fatty acid concentrations in sleep-deprived rats. Concentrations of (**a**) acetic acid, (**b**) propionic acid, (**c**) butyric acid, and (**d**) isobutyric acid. Values are presented as mean ± standard error (n = 6). Different letters indicate significant differences among groups at *p* < 0.05 as per one-way analysis of variance. C—control; SD—sleep deprivation; SDM—sleep deprivation + melatonin.

**Figure 6 ijms-24-16820-f006:**
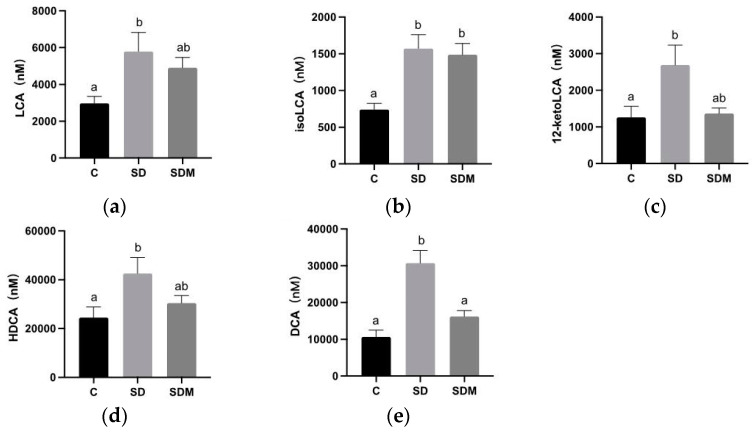
Melatonin administration on fecal secondary bile acid concentrations in sleep-deprived rats. Concentrations of (**a**) lithocholic acid (LCA), (**b**) iso-lithocholic acid (isoLCA), (**c**) 12-keto-lithocholic acid (12ketoLCA), (**d**) hyodeoxycholic acid (HDCA), and (**e**) iso-deoxycholic acid (DCA). Values are presented as mean ± standard error (n = 6). Different letters indicate significant differences among groups at *p* < 0.05 as per one-way analysis of variance. C—control; SD—sleep deprivation; SDM—sleep deprivation + melatonin.

**Figure 7 ijms-24-16820-f007:**
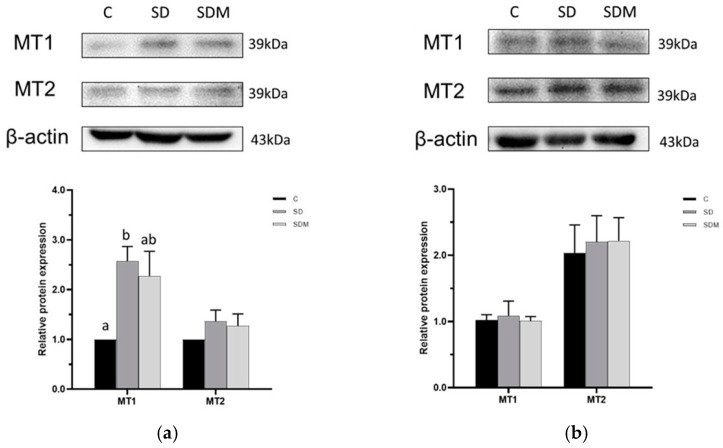
Melatonin administration on melatonin receptors expression levels in cerebral cortex and colon. (**a**) melatonin receptors expression level in the cerebral cortex; (**b**) melatonin receptors expression level in the colon. MT1—melatonin receptors 1; MT2—melatonin receptors 2. Data presented as mean ± standard error (n = 6). Different letters indicated significant differences among groups at *p* < 0.05 as per one-way analysis of variance.

**Table 1 ijms-24-16820-t001:** Effects of melatonin on selected organ weight in sleep-deprived rats.

Groups	C	SD	SDM
**Relative liver weight (%)**	2.49 ± 0.12	2.44 ± 0.10	2.34 ± 0.07
**Relative kidney weight (%)**	0.58 ± 0.01	0.62 ± 0.01	0.60 ± 0.02
**Relative fat weight (%)**			
Perirenal fat	2.67 ± 0.21 ^a^	1.53 ± 0.27 ^b^	1.48 ± 0.24 ^b^
Epididymal fat	1.82 ± 0.17 ^a^	1.15 ± 0.23 ^b^	1.01 ± 0.14 ^b^
**Relative brain weight (%)**			
Whole brain	14.56 ± 0.38	15.72 ± 0.61	15.99 ± 0.62
Hippocampus	1.90 ± 0.17	1.93 ± 0.14	2.24 ± 0.25
Prefrontal cortex	1.23 ± 0.19	1.09 ± 0.14	1.14 ± 0.18
Cerebral cortex	9.49 ± 0.21 ^a^	11.23 ±0.28 ^b^	10.87 ± 0.49 ^ab^
Corpus striatum	1.61 ± 0.10	1.23 ± 0.10	1.57 ± 0.20
Hypothalamus	0.63 ± 0.09	0.65 ± 0.11	0.63 ± 0.04

Relative liver, kidney, perirenal fat, epididymal fat weight and selected brain tissues (whole brain, hippocampus, prefrontal cortex, cerebral cortex, corpus striatum, and hypothalamus). The relative weight of each organ is defined as the original weight of selected organs divided into the individual body weight. Values are presented as the mean ± SEM (n = 6). Different letters indicate significant differences among groups at *p* < 0.05 by one-way ANOVA. C—control; SD—sleep deprivation; SDM—sleep deprivation + melatonin.

**Table 2 ijms-24-16820-t002:** Effects of melatonin on blood biochemical variables in chronically sleep-deprived rats.

Relative Parameters	Group		
	C	SD	SDM
**Serum biochemical variables**			
TG (mg/dL)	114.9 ± 16.6 ^a^	43.1 ± 3.5 ^b^	34.8 ± 3.9 ^b^
TC (mg/dL)	71.0 ± 5.9	73.4 ± 6.0	76.5 ± 7.4
LDL (mg/dL)	5.24 ± 0.59	5.69± 0.82	6.66 ± 0.91
HDL (mg/dL)	22.6 ± 1.1	23.9 ± 1.4	24.1 ± 1.6
FFA (mmol/L)	1.16 ± 0.21 ^a^	0.72 ± 0.08 ^b^	0.65 ± 0.01 ^c^
AST (U/L)	81.9 ± 6.7	75.9 ± 5.4	83.3 ± 8.3
ALT (U/L)	28.9 ± 4.1	25.9 ± 3.4	25.1 ± 2.1
TBIL (mg/dL)	0.04 ± 0.02	0.04 ± 0.01	0.03 ± 0.01
ALB (g/dL)	4.42 ± 0.10 ^a^	4.00 ± 0.22 ^a^	3.58 ± 0.02 ^b^
BUN (mg/dL)	22.3 ± 1.3 ^a^	16.1 ± 1.0 ^b^	16.4 ± 1.1 ^b^
CRE (mg/dL)	0.64 ± 0.06	0.61 ± 0.05	0.57 ± 0.04
Corticosterone (ng/mL)	156.4 ± 24.8 ^a^	275.1 ± 28.9 ^b^	144.8 ± 20.2 ^a^
**Inflammatory status**			
TNF-α (pg/mL serum)	1.66 ± 0.59 ^a^	7.67 ± 1.46 ^b^	7.89 ± 1.48 ^b^
IL-1β (pg/mL serum)	19.8 ± 2.01 ^a^	60.6 ± 16.8 ^b^	69.7 ± 6.4 ^b^
LPS (EU/mL)	0.078 ± 0.021 ^ab^	0.010 ± 0.003 ^b^	0.056 ± 0.002 ^a^
**Neurotransmitter level**			
NGF (pg/100 mL plasma)	1.61 ± 0.10	1.23 ± 0.10	1.57 ± 0.20
Dopamine (pg/100 mL plasma)	0.63 ± 0.09	0.65 ± 0.11	0.63 ± 0.04

Serum triglyceride—TG, total cholesterol—TC, low-density lipoprotein—LDL, high-density lipoprotein—HDL, free fatty acid—FFA, aspartate aminotransferase—AST, alanine aminotransferase—ALT, total bilirubin—TBIL, albumin—ALB, blood urea nitrogen—BUN, and creatinine—CRE levels. Tumor necrosis factor-α—TNF-α, interleukin-1 beta—IL-1β, lipopolysaccharides—LPS, and beta-nerve growth factor—βNGF. Different letters indicated significant differences among groups at *p* < 0.05 by one-way ANOVA. C—control; SD—sleep deprivation; SDM—sleep deprivation + melatonin; Values are presented as the mean ± SEM (n = 6). Different letters indicated significant differences among groups at *p* < 0.05 by one-way ANOVA. C—control; SD—sleep deprivation; SDM—sleep deprivation + melatonin.

**Table 3 ijms-24-16820-t003:** Effects of melatonin on neuropsychiatric behaviors in chronically sleep-deprived rats.

Behavioral Test	Group		
	C	SD	SDM
**Open field test**			
Total distance traveled (m)	6.56 ± 1.78 ^a^	11.26 ± 1.28 ^b^	12.00 ± 1.12 ^b^
Central zone distance traveled (m)	1.16 ± 0.51 ^a^	0.61 ± 0.9 ^b^	0.46 ± 0.07 ^b^
Central zone visit duration (s)	62.3 ± 20.6 ^a^	24.5 ± 3.0 ^b^	29.1 ± 3.4 ^b^
Central zone visit entries	37.9 ± 12.8 ^a^	17.9 ± 1.8 ^b^	16.9 ± 3.8 ^b^
**Elevated plus maze**			
Percentage of open arm duration (%)	48.1 ± 10.1 ^a^	12.9 ± 3.5 ^a^	47.4 ± 9.9 ^a^
Percentage of open arm entries (%)	43.2 ± 8.2 ^a^	25.6 ± 6.0 ^b^	36.0 ± 8.4 ^a^
**Sucrose preference test**			
Week 7 sucrose intake (%)	98.9 ± 1.7	99.1 ± 1.3	99.3 ± 1.1
Week 10 sucrose intake (%)	90.2 ± 3.2 ^a^	71.9 ± 12.1 ^b^	84.2 ± 4.0 ^a^
**Forced swim test**			
Escape to immobility (s)	141.1 ± 16.2	112.6 ± 16.9	135.3 ± 14.2
Total immobility (s)	55.6 ± 10.2 ^a^	89.1 ± 20.2 ^b^	61.1 ± 13.2 ^a^

Total distance traveled, central zone distance traveled, central zone visit duration, and central zone visit entries in the open field test. Total distance traveled, central zone distance traveled, central zone visit duration, and central zone visit entries in the elevated plus maze. Escape to immobility and total immobility in the forced swim test. Values are presented as the mean ± SEM (n = 6). Different letters indicated significant differences among groups at *p* < 0.05 by one-way ANOVA. C—control; SD—sleep deprivation; SDM—sleep deprivation + melatonin.

**Table 4 ijms-24-16820-t004:** Correlation analysis between melatonin receptors expression level and neuropsychiatric behaviors.

**Behavioral Test**	**Cerebral Cortex**		**Colon**		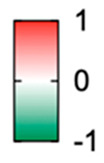
	**MT1**	**MT2**	**MT1**	**MT2**	
**Open field test**					
Total distance traveled (m)	0.78 *	0.21	0.31	0.24	
Central zone visit duration (s)	0.75 *	0.23	0.31	0.23	
Percentage of open arm duration (%)	0.82 *	0.26	0.28	0.17	
**Depression-like behaviors**					
Sucrose preference rate (%)	−0.45 *	0.20	0.26	0.23	
Forced swim test immobility time (s)	0.28	0.17	0.31	0.25	

Pearson’s correlation coefficients (r) are presented. The color changed from dark green (lowest r-value) to dark red (highest r-value). MT1—melatonin receptor 1; MT2—melatonin receptor 2; sucrose preference test; forced swim test. * *p* < 0.05.

## Data Availability

The data used in the study are available from the corresponding author upon reasonable request.

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
