# Peer review of "Melatonin Ameliorates Neuropsychiatric Behaviors, Gut Microbiome, and Microbiota-Derived Metabolites in Rats with Chronic Sleep Deprivation"

_ijms, 2023, doi:10.3390/ijms242316820_

Round 1

Reviewer 1 Report

Comments and Suggestions for Authors

I would like to raise some relatively minor issues.

·         Could different sucrose oral intakes used to detect ‘depression’ have affected some of the alimentary tract content and perhaps other findings?

·         Was the melatonin dose appropriate to provide drug concentrations relevant to the drug’s human therapeutic use? Are you entitled to write of the drug’s intraperitoneal administration as being clinically relevant when its human intake is nearly always oral?

·         In at least 2 different places you write of behavioural responses as being ‘distributed’ through melatonin receptors, etc. By ‘distributed’ do you mean mediated or carried out?

·         You write of ‘relative’ weights but I cannot understand what they are relative to. If it the Control rats, the values for these are percentages and not units of weight.

·         In numbered line 87 of the paper it is stated that there was a ‘single’ melatonin dose. Elsewhere it is repeatedly indicated thar melatonin was administered daily for 3 weeks.

·         I think the use of the ‘a’ and ‘b’ superscripts could be explained more clearly, for example that ‘a’ is used when there is no statistically significant difference between 2 numbers, ‘b’ when there is one.  Also, in the top 2 rows of Table 3, I suspect the numerals 1 and 2 may have been used instead of a and b.

·         In Fig 5, the lower case ‘a’ in the legend seems to appear as ‘A’ in the illustration, and other legend lower case letters be treated similarly.

·         In numbered line 407, does ‘controllable’ mean ‘be controlled for’?

Comments on the Quality of English Language

Dealt with in Comments to the Editor

Author Response

As attached. 

Reviewer 2 Report

Comments and Suggestions for Authors

The work was carried out at a good methodological level, combines both biochemical and neurophysiological studies. Overall, this is a well-designed experiment that allows for a comprehensive assessment of the role of melatonin in SD (sleep deprived) rats. The work is presented in good scientific language and is interesting to read. However, the question arises, why did the authors of the study not measure the average serum concentration of melatonin, at least on the last day of the study, after the rats were sacrificed? Despite the fact that the serum concentration of melatonin fluctuates significantly throughout the day, it would be possible to estimate at a specific point in time the concentration of melatonin in SD rats and SDM (sleep deprived and melatonin-supplemented) rats compared to controls. Without directly measuring the concentration of melatonin itself, it is difficult to draw definitive conclusions about its role in SD rats. Otherwise, all important and interesting evidence becomes indirect. It is not clear from the experiment whether the serum concentration of melatonin increased when it was introduced into SDM, or whether melatonin was metabolized to any analogue. It is also unclear whether melatonin levels decreased in the SD group compared to controls. In any case, if melatonin was not measured directly, it would be necessary to provide literature data showing its active principle.

The essence of the experiment is clearly stated in the materials and methods. Please deposit the raw data on metagenomic sequencing of gut microbiota in GenBank, and provide an accession number in the materials and methods so that readers have access to the information you obtained during the experiment. The question also arises as to why gut microbiota was only examined on the last day of the experiment. It would be highly desirable to track the resulting dramatic changes in the composition of the gut microbiota of the studied groups of rats over time, at least with an interval of 1-2 weeks after the start of the experiment, so that 2-4 time points could be compared. The same question arises in relation to the study of short-chain fatty acids (SCFA) and secondary bile acids (SBA). Why did the experimental design not involve studying their content over time? Since changes in these metabolites are associated with changes in the composition of the gut microbiota, there may have been important fluctuations in the content of these compounds during the experiment.

For a clearer assessment of the significance of the data obtained, enter the Conclusion section, where you briefly formulate the place of your study among similar experiments on the administration of melatonin to reduce the negative effects of SD.

In your work you noted that the potential pathway through which the gut microbiota interacts with the host is through the metabolites of the gut microbiota, such as short-chain fatty acids (SCFAs), secondary bile acids (SBAs), and indoles. You have shown that the composition of gut microbiota and the above metabolites changes significantly in both SD rats and SDM rats compared to controls. However, the authors did not formulate an answer to the question that arises: by what mechanism SD and SDM lead to such a significant change in gut microbiota. This issue should be discussed and added to the Discussion section, since the comparative data you provided on the compositions of gut microbiota of the three studied groups are very different. If there is currently no generally accepted explanation for the relationship between the composition of gut microbiota and SD, then this should also be written in the light of your results.

Author Response

As attached.

Round 2

Reviewer 2 Report

Comments and Suggestions for Authors

The authors responded to most of the comments and made the necessary additions to the article. This makes the work look more holistic. However, the authors, unfortunately, completely ignored my important note, where I asked to deposit in GenBank the raw data that was obtained after metagenomic sequencing of gut microbiota. The authors cited this question along with my other question about the need to do metagenomic sequencing at different time points. And they answered that they did metagenomic sequencing only at the end of the experiment, because they were limited in budget.

However, the sequencing performed by metagenomics was not confirmed either by raw data or tabulated values (in the form of a file in the supplementary). To publish your work, please deposit your data in GenBank and place the accession number in the materials and methods section.

Author Response

International Journal of Molecular Sciences -- MDPI

RE:

Melatonin ameliorated neuropsychiatric behaviors, gut microbiome, and microbiota-derived metabolites in rats with chronic sleep deprivation

Manuscript ID: ijms-2736994

Dear Professor Terézia Kisková, Guest Editor

My co-authors and I appreciate the reviewer’s constructive comments and suggestions. We have revised our manuscript accordingly (responses in bold followed by relevant text excerpts).

Reviewer 2

Q: The authors responded to most of the comments and made the necessary additions to the article. This makes the work look more holistic. However, the authors, unfortunately, completely ignored my important note, where I asked to deposit in GenBank the raw data that was obtained after metagenomic sequencing of gut microbiota. The authors cited this question along with my other question about the need to do metagenomic sequencing at different time points. And they answered that they did metagenomic sequencing only at the end of the experiment, because they were limited in budget.

However, the sequencing performed by metagenomics was not confirmed either by raw data or tabulated values (in the form of a file in the supplementary). To publish your work, please deposit your data in GenBank and place the accession number in the materials and methods section.

Response:

We appreciate your constructive suggestion. Please accept my apology for previous unclear revision. As recommended, we have modified the relevant text and uploaded the original data of 16s ribosomal RNA sequencing of microbiome in Supplementary Materials (ijms--2736994. (Page 15 Line545, 556-557)